# Integrating Remote Sensing and Meteorological Data to Predict Wheat Stripe Rust

Chao Ruan [1,2], Yingying Dong [1,2,*], Wenjiang Huang [1,2,3], Linsheng Huang [4], Huichun Ye [1,2], Huiqin Ma [1,2], Anting Guo [1,2] and Ruiqi Sun [1,2]

1 Key Laboratory of Digital Earth Science, Aerospace Information Research Institute, Chinese Academy of Sciences, Beijing 100094, China; ruanchao@aircas.ac.cn (C.R.); huangwj@aircas.ac.cn (W.H.); yehc@radi.ac.cn (H.Y.); mahq@aircas.ac.cn (H.M.); guoat@aircas.ac.cn (A.G.); sunruiqi19@mails.ucas.ac.cn (R.S.)
2 University of Chinese Academy of Sciences, Beijing 100049, China
3 State Key Laboratory of Remote Sensing Science, Aerospace Information Research Institute, Chinese Academy of Sciences, Beijing 100094, China
4 National Engineering Research Center for Agro-Ecological Big Data Analysis & Application, Anhui University, Hefei 230601, China; 06035@ahu.edu.cn
* Correspondence: dongyy@aircas.ac.cn; Tel.: +86-10-82178178

**Abstract:** Wheat stripe rust poses a serious threat to wheat production. An effective prediction method is important for food security. In this study, we developed a prediction model for wheat stripe rust based on vegetation indices and meteorological features. First, based on time-series Sentinel-2 remote sensing images and meteorological data, wheat phenology (jointing date) was estimated using the harmonic analysis of time-series combined with average cumulative temperature. Then, vegetation indices were extracted based on phenological information. Meteorological features were screened using correlation analysis combined with independent t-test analysis. Finally, a random forest (RF) was used to construct a prediction model for wheat stripe rust. The results showed that the RF model using the input combination (phenological information-based vegetation indices and meteorological features) produced a higher prediction accuracy and a kappa coefficient of 88.7% and 0.772, respectively. The prediction model using phenological information-based vegetation indices outperformed the prediction model using single-date image-based vegetation indices, and the overall accuracy improved from 62.9% to 78.4%. These results indicated that the method combining phenological information-based vegetation indices and meteorological features can be used for wheat stripe rust prediction. The results of the prediction model can provide guidance and suggestions for disease prevention in the study area.

**Keywords:** wheat stripe rust; prediction; vegetation indices; meteorological feature; phenology

## 1. Introduction

Wheat is one of the three major food crops in China, and its production is related to the country's food security [1]. Stripe rust is an epidemic disease caused by *Puccinia striiformis* f. sp. *tritici* (Pst), which seriously affects the yield and quality of wheat [2]. In epidemic years, wheat stripe rust can cause more than 30% yield loss or even result in extermination if untimely prevention and favorable weather conditions occur [2,3]. In China, the average annual area of stripe rust is over 4 million hectares, with an average annual yield loss of approximately 1 million tons [4,5]. Therefore, the prevention and control of wheat stripe rust are crucial to ensuring food security. Predicting the occurrence of the disease with high accuracy on a regional scale is the key to prevention and control and can provide timely and reliable management advice to agricultural plant protection departments to minimize losses [6].

There are two main methods of predicting crop diseases. The first method primarily uses meteorological data to predict the occurrence of diseases. Because spore survival, germination and infestation require suitable habitat conditions, some scholars have predicted the disease occurrence trend by analyzing the response relationship between habitat conditions and diseases by using statistical models [7,8]. The second method is to use remote sensing data for disease prediction. Diseases can change crop growth conditions such as pigment content, water content, biomass and cell structure [2,3,9]. In general, these crop growth conditions can be characterized by vegetation indices calculated from remote sensing images [10–12]. Therefore, many scholars have constructed prediction models and predicted the spatial distribution for different diseases by selecting vegetation indices that can characterize crop growth conditions under disease stress [13,14]. For example, Ruan et al. used a sequence forward selection algorithm to select vegetation indices sensitive to stripe rust from Sentinel-2 remote sensing images, combined with the support vector machine (SVM), successfully constructed a prediction model and predicted the distribution of wheat stripe rust occurrence in different periods [15]. Although these methods, which account for habitat and crop growth conditions, respectively, have been proven to be useful for crop disease prediction, some new methods have been proposed to improve the accuracy of prediction models.

In current state-of-the-art research, some scholars have constructed crop disease prediction models by combining crop growth and habitat conditions, which have been shown to be effective in improving prediction model accuracy [16]. For example, Zhang et al. used a series of vegetation indices and meteorological features extracted from HJ remote sensing images and meteorological data, combined with a logistic method to construct a wheat powdery mildew prediction model [17]. Xiao et al. successfully constructed a prediction model for wheat scab based on the vegetation indices and meteorological characteristics using Landsat 8 remote sensing images and meteorological data [18]. These studies show that, compared with prediction models based on a single type of feature, the input combination of vegetation indices and meteorological features can effectively improve the accuracy of the prediction model [19]. These results motivate us to continue to explore the prediction methods of wheat stripe rust by combining crop growth and habitat conditions. In actual regional field management, wheat phenology varies in the same period due to climate and some agronomic factors [20]. Changes in crop growth conditions caused by phenological differences can cause greater interference with changes in crop growth conditions caused by wheat stripe rust stress. For example, as wheat grows, growth conditions such as chlorophyll content and biomass gradually increase from the regreening to the heading stage [21]. These changes lead to a corresponding change in the vegetation indices, which characterizes wheat growth conditions [22,23]. However, the majority of existing studies used remote sensing images of a single date combined with meteorological data to predict diseases, ignoring the impact of phenological differences on crop disease prediction [5,17,18].

In this study, we focused on predicting wheat stripe rust using phenological information-based vegetation indices and meteorological features. The main objectives were (1) to extract vegetation indices based on phenological information for wheat stripe rust prediction, (2) to compare and evaluate the feasibility of combining vegetation indices and meteorological features to predict wheat stripe rust and (3) to predict and map wheat stripe rust at the study site using an optimal prediction model.

## 2. Materials and Methods

### 2.1. Field Survey and Data Collection

#### 2.1.1. Study Site

The study site is located in Qishan County, Baoji City, Shaanxi Province, China (34°18′–34°30′N, 107°33′–107°47′E) (Figure 1). In this region, the main soil types of agricultural land are lou soil, loessial soil and fluvo-aquic soil [24]. Wheat is the main crop, and the main variety is Xinong 822, which is susceptible to stripe rust. The county is located in

the Guanzhong Plain of Shaanxi Province, which is the border area between the overwintering and spring epidemic areas of wheat stripe rust in China [25]. The average annual temperature and precipitation of the county are 6–13 °C and 500–700 mm, respectively [26]; low-temperature and high-humidity environmental conditions increase the risk of stripe rust [27]. Stripe rust is the major wheat disease in the region, and it occurred severely in 2021.

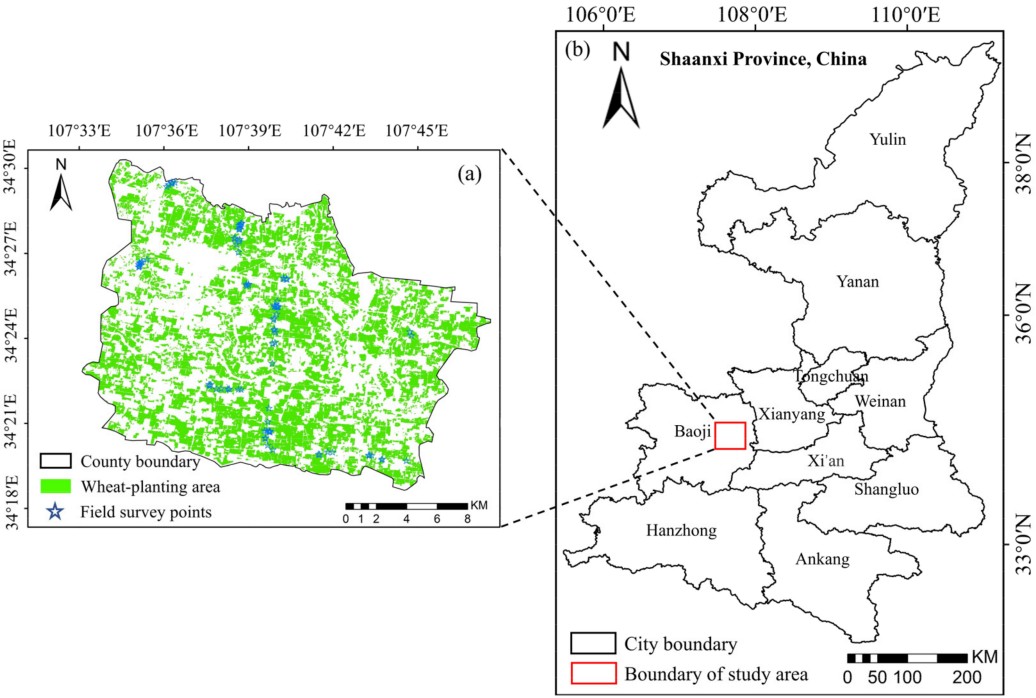

**Figure 1.** Geographic location of the study site and spatial distribution of field survey points and wheat-planting areas. (**a**) The distribution of field survey sites and wheat-planting areas in the study site; and (**b**) the location of the study site in Shaanxi Province.

### 2.1.2. Wheat Stripe Rust Survey Data Acquisition

In the field survey (17 April 2021), a total of 97 plots were investigated. To match the plot and remote sensing image pixels, a 10 m × 10 m plot was selected within a 20 m × 20 m region, where wheat growth was relatively uniform, and the severity of stripe rust was surveyed. Five 1 m × 1 m quadrats were selected at the four corners and center of each plot, and ten leaves were randomly selected in each quadrat to record disease severity. The average severity of the five quadrats was used to represent the severity of the plot. The center latitude and longitude of each plot were recorded by a submeter-precision handheld global positioning system (GPS) receiver. The severity of stripe rust was measured according to the rule of the National Rules for the Investigation and Forecasting of Crop Diseases (GB/T 15795-2011) [28]. Due to the generally mild occurrence of stripe rust at the study site, the severity of the plot was classified into two categories, healthy and diseased, for subsequent studies.

### 2.1.3. Remote Sensing Data Acquisition and Preprocessing

Sentinel-2 remote sensing images from January to June 2018–2021 (wheat-growing season) were downloaded from the European Space Agency Sentinels Scientific Data Hub (https://scihub.copernicus.eu/ (accessed on 22 January 2022)) for the study site. The Sentinel-2 remote sensing images were preprocessed for atmospheric corrections using the Sen2cor module and resampled to 10 m using the resampling tool in the Sentinel Application Platform software (SNAP). The decision tree method combined with multitemporal phenological information was used to extract the spatial distribution of wheat at the

study site [29–31]. Ninety-seven field survey plots were used to verify the accuracy of the extracted wheat-planting area, and they showed an overall accuracy of 98%.

### 2.1.4. Meteorological Data Acquisition and Preprocessing

The daily meteorological data were obtained for June 2018–2021 and December 2020 from the National Meteorological Information Center in 42 national benchmark weather stations in Sichuan province, Gansu province and Chongqing city adjacent to Qishan county. The inverse distance weighting method in the ArcGIS software was used to spatially interpolate the meteorological data for subsequent continuous spatial pixel-scale analysis [18]. Leave-one-out cross-validation was applied to verify the interpolation accuracy. Each weather station was used as a validation sample and the other weather stations for spatial interpolation. The $R^2$ between the interpolation results and true meteorological value was used as the fitting accuracy, and the fitting accuracy was greater than 0.9. The data included nine types of meteorological elements: Average temperature (ATEM), maximum temperature (HTEM), minimum temperature (LTEM), average ground temperature (AGST), maximum ground temperature (HGST), minimum ground temperature (LGST), sunshine hours (SSD), precipitation (PRE) and relative humidity (RHU).

### 2.2. Construction of the Prediction Model for Wheat Stripe Rust

In this study, the prediction model for wheat stripe rust was constructed in three steps (Figure 2). First, we estimated the wheat phenology (jointing date) based on time-series remote sensing images and meteorological data using the harmonic analysis of time-series combined with average cumulative temperature and extracted vegetation indices based on the estimation results to characterize wheat growth conditions. Then, meteorological features sensitive to stripe rust were screened as habitat conditions by correlation analysis combined with independent t-test analysis. Finally, the phenological information-based vegetation indices and meteorological features were used as model inputs, and a prediction model for wheat stripe rust was constructed using the random forest (RF) method.

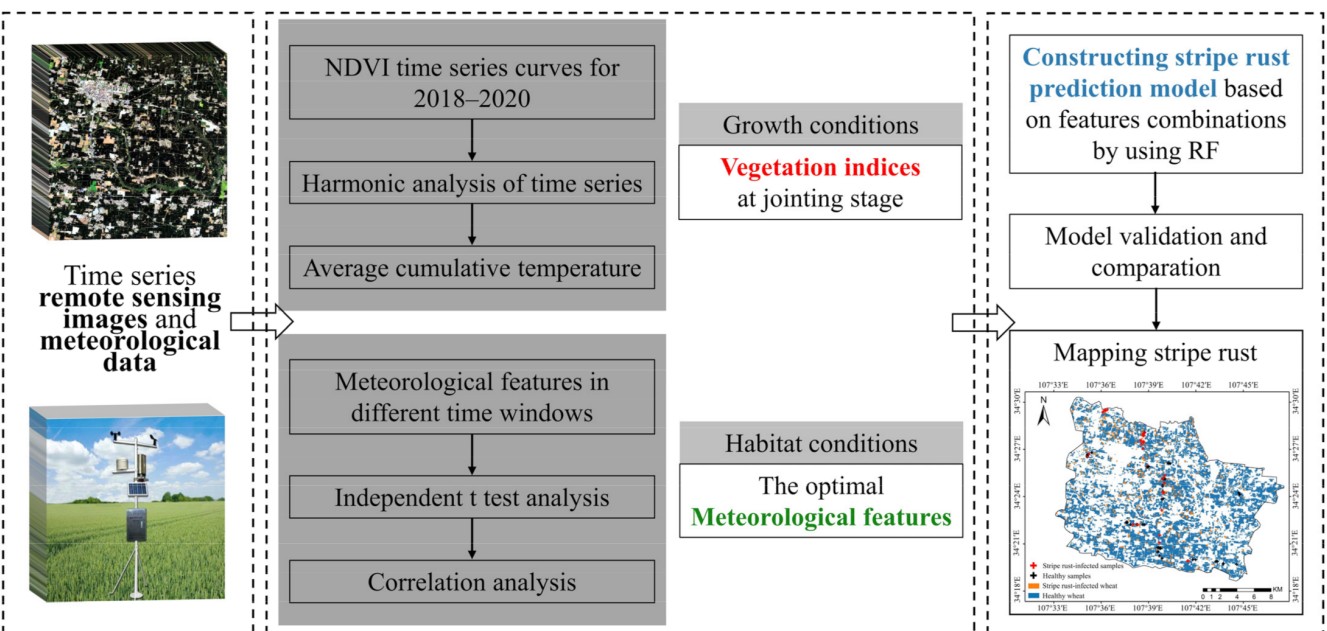

**Figure 2.** Flowchart of the prediction model for wheat stripe rust.

### 2.2.1. Growth Conditions Extraction

On the one hand, physiological and biochemical parameters that reflect the growth conditions of wheat, such as biomass, pigment content and moisture content, change after stripe rust infestation [5]. These changes can be characterized by vegetation indices

consisting of multiple sensitive spectral bands [12]. On the other hand, the jointing stage is critical phenology when wheat stripe rust starts to develop rapidly [12]. During this stage, stripe rust fungi rely on wheat nutrients and water to proliferate and destroy cell structures [2,3]. Therefore, this study extracted phenological information-based vegetation indices to characterize wheat growth conditions for predicting the occurrence of wheat stripe rust. Three key steps were included as follows:

Vegetation Indices Extraction

Based on the previous results of the research team, a total of 6 vegetation indices sensitive to wheat stripe rust at the jointing stage were selected to characterize the wheat growth conditions. These vegetation indices include the normalized difference vegetation index (NDVI), red-edge disease stress index (REDSI), disease water stress index (DSWI), normalized difference vegetation index red-edge (NDVIre), triangular vegetation index (TVI) and plant senescence reflectance index (PSRI) (Table 1) [15]. Among them, NDVI and NDVIre can be used to characterize crop biomass and as an indicator of nitrogen [32–35], DSWI can be used to capture crop water status [17], REDSI is associated with stripe rust stress [28], PSRI is associated with crop pigment content and health status [36] and TVI monitors chlorophyll content [37].

**Table 1.** Vegetation indices sensitive to wheat stripe rust.

| Vegetation Indices Definition | Formula | Correlation |
|---|---|---|
| Normalized Difference Vegetation Index, NDVI [38] | $(R_{NIR} - R_R)/(R_{NIR} + R_R)$ | LAI, biomass |
| Normalized Difference Vegetation Index red-edge, NDVIre [39] | $(R_{NIR} - R_{Re2})/(R_{NIR} + R_{Re2})$ | LAI, biomass |
| Plant Senescence Reflectance Index, PSRI [40] | $(R_R - R_G)/R_{Re2}$ | Pigment content, Vegetation health |
| Red-edge Disease Stress Index, REDSI [28] | $((705 - 665)(R_{Re3} - R_R) - (783 - 665)(R_{Re1} - R_R))/(2R_R)$ | Sensitive to stripe rust |
| Triangular Vegetation Index, TVI [41] | $0.5[120(R_{NIR} - R_G) - 200(R_R - R_G)]$ | Vegetation status |
| Disease Water Stress Index, DSWI [42] | $(R_{NIR} + R_G)/(R_{SWIR} + R_R)$ | Water status |

The Wheat-Jointing Date Estimation

In this study, the harmonic analysis of time series (HANTS) combined with the average cumulative temperature was used to estimate the date of wheat jointing [43–45]:

i.   S-G filtering was used to smooth the NDVI time-series images from 2018 to 2020. Then, the smoothed NDVI time-series curves were fitted using HANTS.

ii.  Based on the growth characteristics of wheat, regreening is the stage of wheat growth recovery, corresponding to the date when the NDVI time-series curve reaches its minimum (i.e., the regreening date corresponding to the square in Figure 3), and jointing is the stage of rapid wheat growth, corresponding to the date when the first-order derivatives of the NDVI time-series curve reach maximum (i.e., the jointing date corresponding to the circle in Figure 3) [45–48]. The fitted NDVI curves were used to extract the regreening and jointing dates for 2018–2020 and the regreening date for 2021.

iii. The calculated average cumulative temperature from 2018 to 2020 from the regreening date to the jointing date was used as a threshold to estimate the wheat-jointing date in 2021 based on meteorological data.

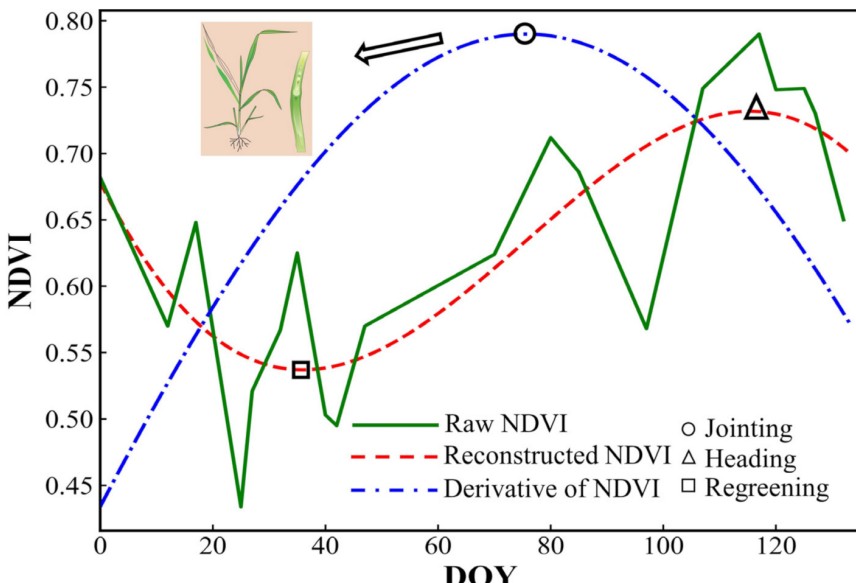

**Figure 3.** Extraction of Day of Year (DOY) of regreening and jointing date according to the NDVI time-series curve. The green solid line indicates the raw NDVI time-series curve. The red dotted line indicates the reconstructed NDVI time-series curve. The blue dotted line indicates the first-order derivative of the NDVI time-series curve. The square indicates the regreening date. The circle indicates the jointing date and the triangle indicates the heading date.

Phenological Information-Based Vegetation Indices Extraction

First, we selected time-series Sentinel-2 remote sensing images within the time range of the jointing date at the study site and calculated the six selected vegetation indices (Table 1). Then, the vegetation indices corresponding to the jointing date of each pixel were extracted. In this case, for some pixels without corresponding images from the jointing dates, the images closest to the date were selected to extract the vegetation indices. Figure 4 shows the process of extracting phenological information-based vegetation indices.

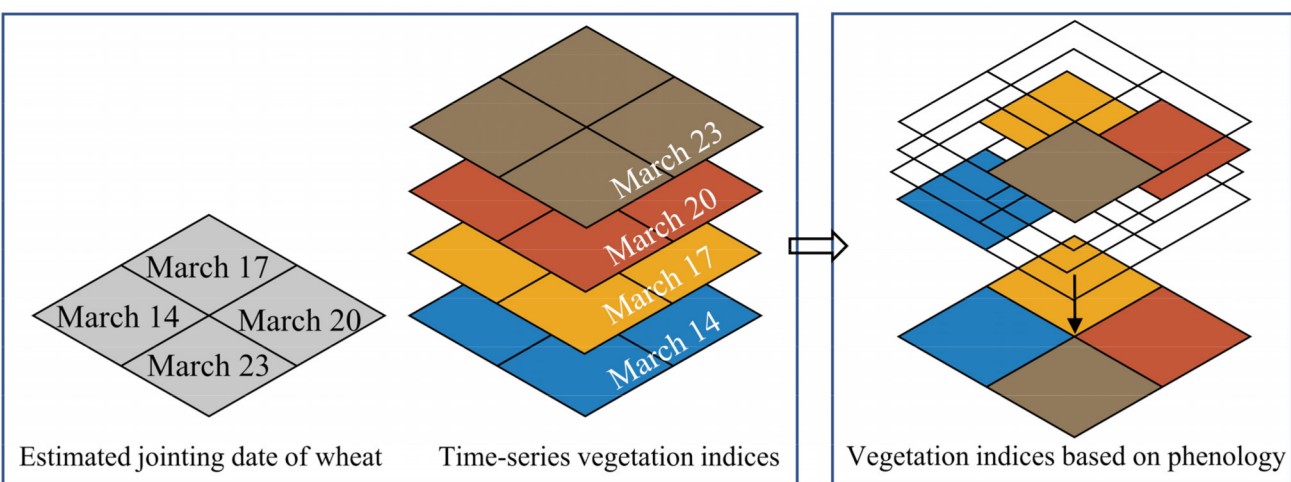

**Figure 4.** The process of extracting phenological information-based vegetation indices.

To compare the differences in response to stripe rust between the phenological information-based vegetation indices and the single-date image-based vegetation indices, the date with the largest proportion of all pixels in the estimated results of the jointing date was used as the single date. The remote sensing image of a particular date was used to calculate the 6 selected vegetation indices to characterize the wheat growth conditions on a single date.

### 2.2.2. Habitat Conditions Extraction

The survival, germination and infection of stripe rust fungi require suitable habitat conditions [49]. On the one hand, the habitat conditions of wheat overwintering (December–February) affect the survival and the initial number of fungi during the spring epidemic of the disease [2,50–52]. On the other hand, under adequate sources and suitable host conditions, the habitat conditions prior to the disease onset determine the reproduction and infestation of stripe rust [53]. Therefore, the extraction of habitat conditions includes two parts. First, we selected meteorological data from December to February and calculated monthly averages of nine types of meteorological elements for each month (a total of 27 meteorological features) to characterize the overwintering habitat conditions of the wheat. Then, the mean values of nine types of meteorological elements were calculated under four windows (a total of 36 meteorological features) to characterize the pre-infestation habitat conditions, using the field survey date as the ending time (17 April) and 7 days (10 April to 16 April), 14 days (3 April to 16 April), 21 days (27 March to 16 April) and 28 days (20 March to 16 April) ahead of this time as time windows. Finally, a total of 63 meteorological features were calculated as candidate features.

Although the above meteorological features may be associated with the occurrence of stripe rust, the possible redundancy between these features and the correlation between vegetation indices and stripe rust can affect the accuracy of the prediction model. To eliminate these effects and improve the prediction accuracy of the model, the candidate features were screened by correlation analysis combined with independent t-test analysis to identify the optimal meteorological features. First, the correlation between candidate features and stripe rust was calculated using independent t-test analysis [54]. The features showing statistical significance ($p$-value $< 0.001$) were screened by testing the variability between healthy and stripe rust samples. Then, the correlation coefficient (R) between features was calculated using correlation analysis, and features with an absolute R-value greater than 0.9 were removed. The final retained meteorological features were used to characterize habitat conditions.

### 2.2.3. Wheat Stripe Rust Prediction Model Construction

The prediction model for wheat stripe rust was constructed by inputting phenological information-based vegetation indices and meteorological features (PIVIs + MFs) into the classification method. The occurrence of wheat stripe rust was predicted in each pixel, and the distribution was mapped by the prediction model. Among them, vegetation indices' extraction depends on the update time of remote sensing images; for meteorological features, although their time range is fixed, the meteorological data are actually predictable and become more accurate as the update time approaches. Therefore, the prediction of stripe rust can be achieved as long as remote sensing images and meteorological prediction data are acquired.

The RF method has better performance in remote sensing image classification due to the high classification accuracy and is often used in the construction of prediction models for crop disease [55–57]. Therefore, in this study, the RF method was chosen to construct a prediction model for wheat stripe rust. The RF method is an ensemble classifier, and the main principle is to generate multiple decision trees using randomly selected training samples and subsets of variables [58]. The final classification decision is made by averaging the probabilities of category assignment computed over all the resulting trees [58]. In this study, a grid search method (GS) was used to determine the optimal number of trees [59].

To investigate the feasibility of predicting wheat stripe rust based on PIVIs + MFs, single-date image-based vegetation indices (VIs), phenological information-based vegetation indices (PIVIs) and single-date image-based vegetation indices combined with meteorological features (VIs + MFs) were used as input features to construct prediction models. To validate the usefulness of the RF method in prediction models for stripe rust, the SVM and logistic methods were used to construct the prediction models based on the four types of input features (VIs, PIVIs, VIs + MFs and PIVIs + MFs). The SVM is a

classical machine learning method to maximize the distance between two types of samples by finding the optimal decision boundary [60]. The logistic is the method traditionally used in epidemiological studies [61]. The RF and SVM methods were conducted using MATLAB 2018a software. The logistic method was conducted using the Statistical Package for the Social Sciences (SPSS 26.0). Four model evaluation metrics, namely, overall accuracy (OA), producer accuracy (PA), user accuracy (UA) and kappa coefficient, were used to evaluate the performance of the prediction models [62]. The OA can directly reflect the proportion of correct predictions [63]. PA and UA indicate the prediction accuracy of each category [63]. The kappa coefficient is a measure of the consistency between the predicted and actual results [64]. K-fold cross-validation was used for the training and validation of the prediction model, where k = 10 [65]. The method divides 97 samples into 10 folds, 9 of which are used as the model training set, and the remaining fold is used as the validation set. Then, 10 iterations were performed so that each fold of samples was used as the validation set. Finally, the average results of the evaluation metrics (OA, UA, PA, and kappa) were calculated to represent the performance of the prediction model.

## 3. Results and Discussion

### 3.1. Vegetation Indices for Wheat Stripe Rust

The estimated wheat-jointing dates of the study site in 2021 are shown in Figure 5. The responses of the phenological information-based vegetation indices and the single-date image-based vegetation indices (calculated by remote sensing images of 17 March) are shown in Figure 6, where their normalized mean and standard deviation are compared for healthy and stripe-rust-infected wheat. The results showed that the six phenological information-based vegetation indices differed significantly between healthy and stripe rust samples. The largest difference in the PSRI was mainly due to the destruction of the wheat cell structure by stripe rust fungi, which resulted in red-edge spectral displacement [66,67]. These results are consistent with the results of our team in the extraction of vegetation indices that are sensitive to stripe rust [15]. For the single-date image-based vegetation indices, the differences among PSRI, NDVI and TVI were small.

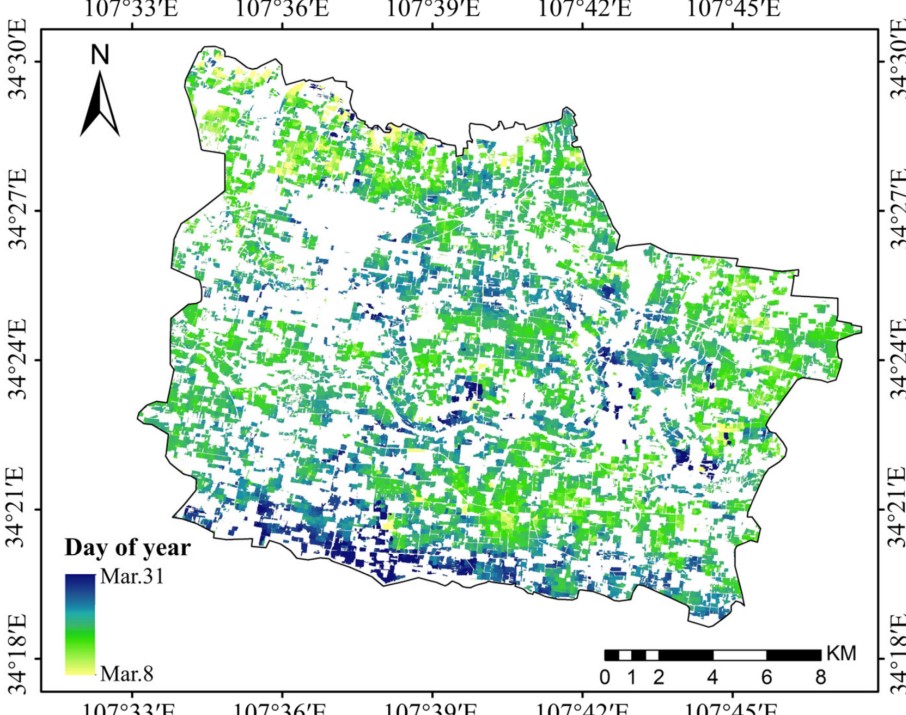

**Figure 5.** Estimation results of wheat-jointing date.

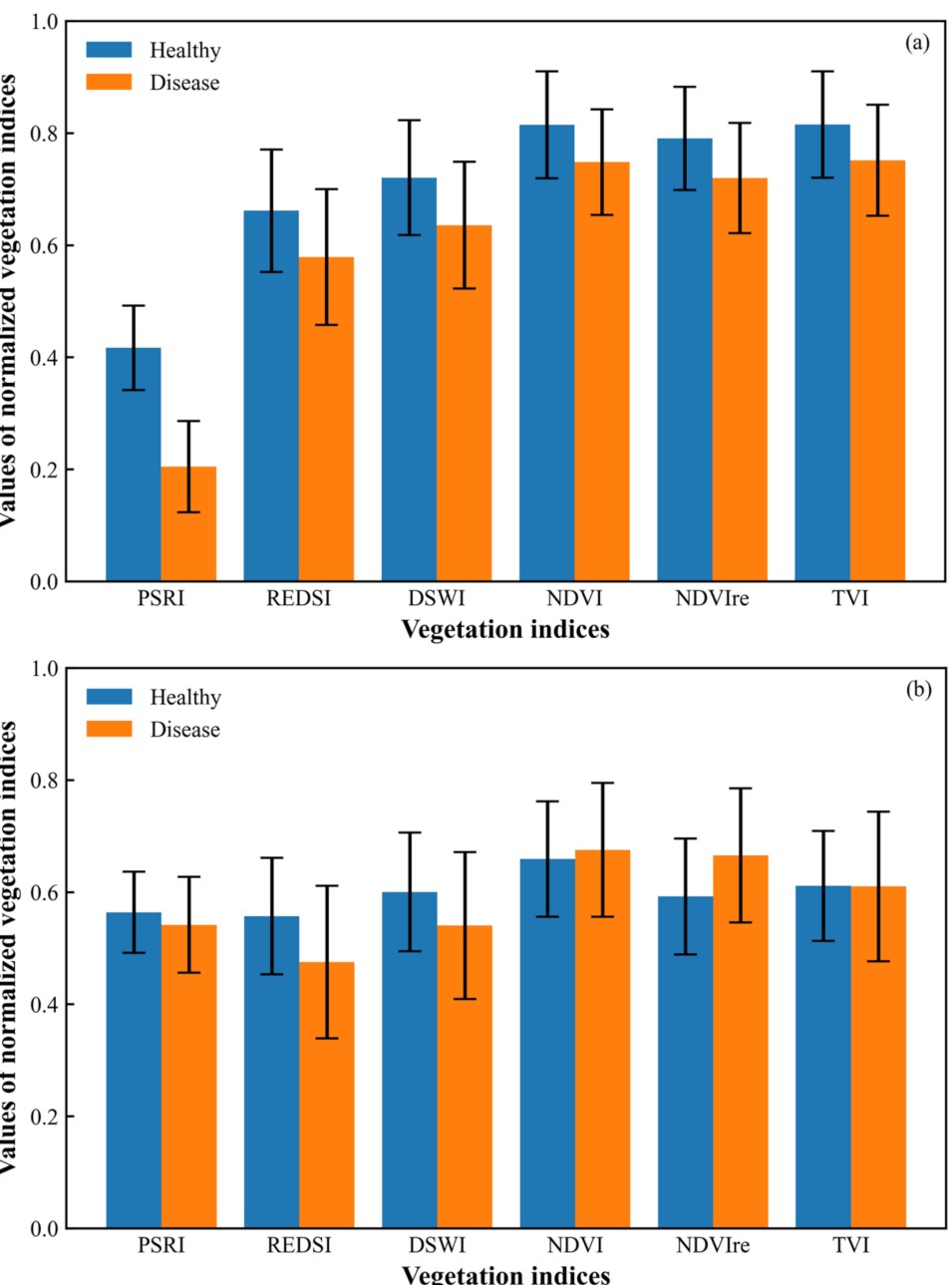

**Figure 6.** The mean and standard deviation of normalized vegetation indices. (**a**) Phenological information-based vegetation indices; and (**b**) single-date image-based vegetation indices. Blue bars indicate the mean values of healthy samples, orange bars indicate the mean values of stripe-rust-infected samples and error bars indicate the standard deviation.

For the two types of vegetation indices, the PSRI, REDSI, and DWSI showed that the mean values of healthy samples were greater than those of stripe-rust-infected samples, and the differences were more pronounced for the phenological information-based vegetation indices. In addition, the phenological information-based NDVI, NDVIre and TVI showed higher normalized values for healthy samples than for stripe-rust-infected samples, while a single-date image-based NDVI and NDVIre showed the opposite trend and the TVI was not significantly different. Generally, for healthy wheat, the NDVI tends to increase between the regreening and heading stages [68], while for stripe-rust-infected wheat, the NDVI decreases due to the uptake of wheat nutrients and the reduction of wheat biomass by stripe rust fungi [69]. In addition, wheat phenology differs in the same period due to sowing

time, climate and other factors [20]. Therefore, phenological information-based vegetation indices can effectively eliminate the interference of wheat phenological differences on stripe rust. For the single-date image-based vegetation indices, the overall difference was not significant because it ignored the phenological differences. These may be the main reasons for the differences in the response of the two types of vegetation indices to stripe rust. These results suggest that phenological information-based vegetation indices can better characterize wheat growth conditions under stripe rust stress and are more suitable for stripe rust prediction.

### 3.2. Meteorological Features for Wheat Stripe Rust

To identify meteorological features that are sensitive to stripe rust, 63 candidate meteorological features extracted based on the meteorological interpolation results were used for feature extraction. Figure 7 shows the interpolation results of the meteorological data by taking the average temperature and precipitation in December 2020 as examples. First, 17 meteorological features that showed statistical significance (*p*-value < 0.001) were screened using independent t-test analysis, including 12 overwintering and 5 spring epidemic meteorological features (Table 2). Notably, there were significantly more meteorological features screened for the overwintering epidemic than for the spring epidemic. In our case study, mid-April was at the beginning of the stripe rust epidemic at the study site, and under suitable climatic conditions, the occurrence of stripe rust was mainly related to the number of overwintering fungi [70]. The number of fungi is largely influenced by habitat conditions during overwintering [70]. This may be the main reason for the above results.

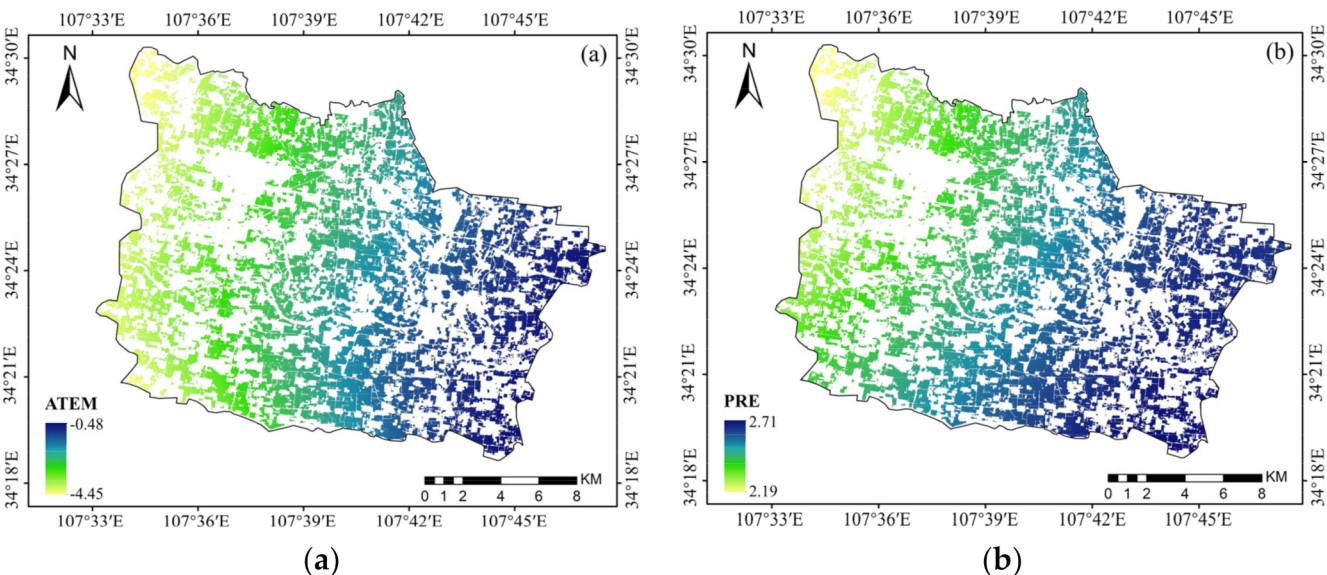

**Figure 7.** The interpolation results of meteorological data in Qishan county. (**a**) Average temperature (ATEM; 0.1 °C) in December 2020; (**b**) average precipitation (PRE; 0.1 mm) in December 2020.

To further reduce the redundancy between the 17 meteorological features mentioned above, the correlation coefficient (R) between the meteorological features was calculated using correlation analysis (Figure 8). The results show that there was a high redundancy between the features. Among them, there were high correlations among ATEM, PRE and RHU in the same time windows. There were high positive correlations between TEM_M01 and PRE_M01 and high negative correlations with RHU_M01 TEM_M02, PRE_M02 and RHU_M01. An absolute value of R greater than 0.9 was used as the threshold, and four meteorological features sensitive to stripe rust were finally screened, namely, average relative humidity in December (RHU_M12), average sunshine hours in January (SSD_M01), average precipitation in January (PRE_M01) and average precipitation 7 days before the field survey date (PRE_B07).

**Table 2.** Significance of 63 candidate features calculated using independent t-test analysis.

| Feature | Time Window | | | | | | |
|---|---|---|---|---|---|---|---|
| | **M12** | **M01** | **M02** | **B07** | **B14** | **B21** | **B28** |
| ATEM | | *** | *** | | | | |
| HTEM | *** | | | | | | |
| LTEM | | *** | *** | | | | |
| AGST | | | | | | | |
| HGST | | | | | | | *** |
| LGST | | *** | | | | | |
| SSD | | *** | | *** | | *** | *** |
| PRE | | *** | *** | *** | | | |
| RHU | *** | *** | *** | | | | |

Note: M12, M01 and M02 represent December 2020, January 2021 and February 2021, respectively; B07, B14, B21 and B28 represent 7, 14, 21 and 28 days before the field survey date, respectively; *** represents healthy wheat and stripe-rust-infested wheat significantly different at the confidence level of 0.999 (*p*-value < 0.001).

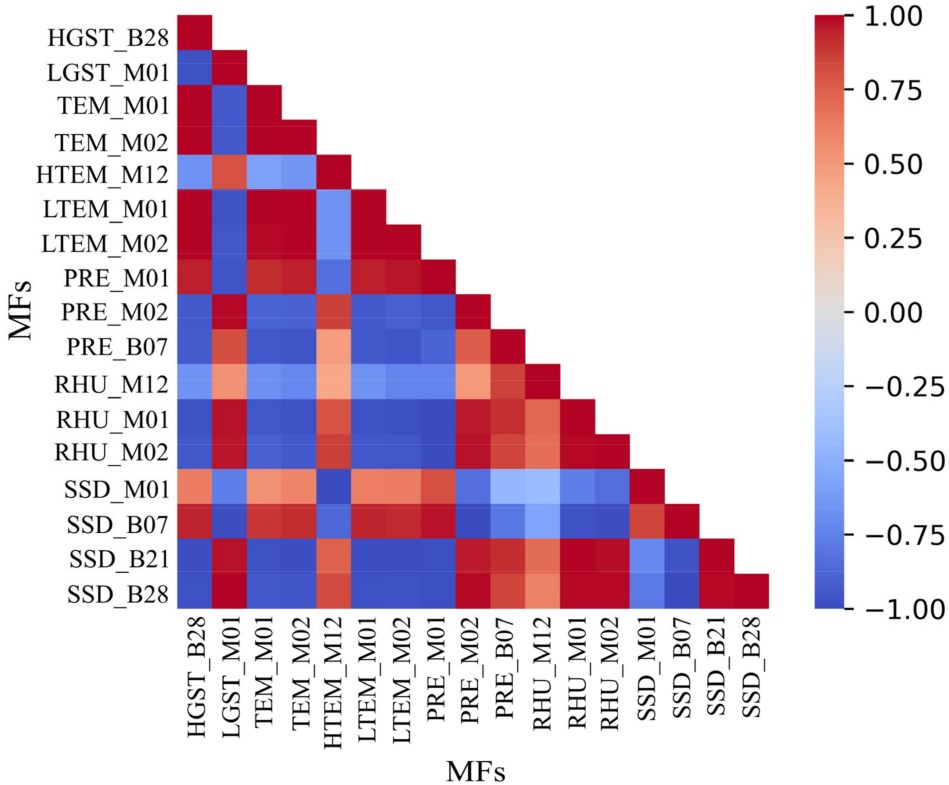

**Figure 8.** Heatmap of correlation coefficients between meteorological features calculated using correlation analysis. Red represents a positive correlation between features, blue represents a negative correlation between features and the darker the color is, the higher the correlation between features.

*3.3. Evaluation of Prediction Model for Wheat Stripe Rust*

The RF method was used to construct the prediction model for wheat stripe rust by combining phenological information-based vegetation indices and meteorological features (PIVIs + MFs) as input features. To evaluate and compare the performance of the prediction model using PIVIs + MFs, VIs, PIVIs and VIs + MFs were also used to construct the RF prediction models. The parameters of the prediction models are shown in Table 3. Table 4 shows the confusion matrix for the predicted results of each model. The results showed that the RF prediction models constructed using vegetation indices combined with meteorological features (VIs + MFs and PIVIs + MFs) were better than those constructed using vegetation indices (VIs and PIVIs), with an overall improvement in accuracy of 10.3% and 9.3% and an improvement in kappa coefficients of 0.206 and 0.186, respectively.

These results suggest that combining vegetation indices and meteorological features for stripe rust prediction can improve prediction model accuracy. These results are consistent with the findings of Xiao et al. and Ma et al. in the prediction of wheat scab and wheat powdery mildew [18,71]. In addition, the overall accuracy and kappa coefficient of the RF prediction model constructed using PIVIs outperformed those constructed using VIs by 7.3% and 0.144, respectively. The overall accuracy and kappa coefficient of the RF prediction model constructed using PIVIs + MFs outperformed those constructed using VIs + MFs by 8.3% and 0.164, respectively. These results indicate that phenological information-based vegetation indices can effectively eliminate the influence of phenological differences on stripe rust prediction and improve the accuracy of the prediction model.

**Table 3.** The model parameters of the random forest (RF) and support vector machine (SVM).

| Feature | Parameter of RF | Parameter of SVM | |
| --- | --- | --- | --- |
| | The Number of Trees | C | $\gamma$ |
| VIs | 3 | 5.31 | 12.18 |
| PIVIs | 3 | 2.87 | 13.67 |
| VIs + MFs | 5 | 1.58 | 13.91 |
| PIVIs + MFs | 5 | 0.88 | 8.23 |

**Table 4.** Confusion matrix for the RF prediction model.

| Feature | | Healthy | Infected | Sum | UA | OA | Kappa |
| --- | --- | --- | --- | --- | --- | --- | --- |
| VIs | Healthy | 36 | 13 | 49 | 73.5% | | |
| | Infected | 15 | 33 | 48 | 68.8% | | |
| | Sum | 51 | 46 | 97 | | 71.1% | 0.422 |
| | PA | 70.6% | 71.7% | | | | |
| PIVIs | Healthy | 40 | 10 | 50 | 80% | | |
| | Infected | 11 | 36 | 47 | 76.6% | | |
| | Sum | 51 | 46 | 97 | | 78.4% | 0.566 |
| | PA | 78.4% | 78.3% | | | | |
| VIs + MFs | Healthy | 41 | 9 | 50 | 82% | | |
| | Infected | 10 | 37 | 47 | 78.7% | | |
| | Sum | 51 | 46 | 97 | | 80.4% | 0.608 |
| | PA | 80.4% | 80.4% | | | | |
| PIVIs + MFs | Healthy | 45 | 5 | 50 | 90% | | |
| | Infected | 6 | 41 | 47 | 87.2% | | |
| | Sum | 51 | 46 | 97 | | 88.7% | 0.772 |
| | PA | 88.2% | 89.1% | | | | |

To evaluate and compare the performance of the prediction model based on RF methods, the SVM and logistic method were used to construct the prediction models using four feature sets (VIs, PIVIs, VIs + MFs and PIVIs + MFs). The confusion matrix of the prediction results is shown in Tables 5 and 6. The results showed that the SVM and logistic prediction model using the four feature sets performed consistently with the RF prediction model (Table 4). That is, a combination of vegetation indices and meteorological features outperformed a single type of feature in wheat stripe rust prediction. The phenological information-based vegetation indices performed better than the single-date image-based vegetation indices. Moreover, the prediction model using PIVIs + MFs was the best among all models. However, the RF prediction models all outperformed the SVM and logistic prediction models for the same feature inputs, with overall accuracies of 4.1% and 8.2% (VIs), 4.2% and 8.3% (PIVIs), 4.1% and 6.2% (VIs + MFs) and 3.1% and 7.3% (PIVIs + MFs), respectively. Meanwhile, the SVM prediction models all outperformed the logistic prediction models for the same feature inputs, with overall accuracies of 4.1% (VIs), 4.1% (PIVIs), 2.1% (VIs + MFs) and 4.2% (PIVIs + MFs). These results are consistent with

the results of Li et al. in the extraction of the forest type, Tang et al. in the prediction of wheat aphids and Tien Bui et al. in landslide prediction [72–74]. A possible reason for this is the excellent performance of the RF and SVM with the RBF as the kernel function in the linearly inseparable case [75], which was able to capture the nonlinear relationship between stripe rust and vegetation indices and meteorological features [76]. Moreover, the RF method can successfully handle high-dimensional data [55].

**Table 5.** Confusion matrix for the SVM prediction model.

| Feature | | Healthy | Infected | Sum | UA | OA | Kappa |
|---|---|---|---|---|---|---|---|
| VIs | Healthy | 34 | 15 | 49 | 69.4% | | |
| | Infected | 17 | 31 | 48 | 64.5% | | |
| | Sum | 51 | 46 | 97 | | 67.0% | 0.340 |
| | PA | 66.7% | 67.4% | | | | |
| PIVIs | Healthy | 38 | 12 | 50 | 76% | | |
| | Infected | 13 | 34 | 47 | 72.3% | | |
| | Sum | 51 | 46 | 97 | | 74.2% | 0.484 |
| | PA | 74.5% | 73.9% | | | | |
| VIs + MFs | Healthy | 39 | 11 | 50 | 78% | | |
| | Infected | 12 | 35 | 47 | 74.5% | | |
| | Sum | 51 | 46 | 97 | | 76.3% | 0.525 |
| | PA | 76.5% | 76.1% | | | | |
| PIVIs + MFs | Healthy | 44 | 7 | 51 | 86.3% | | |
| | Infected | 7 | 39 | 46 | 84.8% | | |
| | Sum | 51 | 46 | 97 | | 85.6% | 0.711 |
| | PA | 86.3% | 84.8% | | | | |

**Table 6.** Confusion matrix for the logistic prediction model.

| Feature | | Healthy | Infected | Sum | UA | OA | Kappa |
|---|---|---|---|---|---|---|---|
| VIs | Healthy | 32 | 17 | 49 | 65.3% | | |
| | Infected | 19 | 29 | 48 | 60.4% | | |
| | Sum | 51 | 46 | 97 | | 62.9% | 0.257 |
| | PA | 62.7% | 63% | | | | |
| PIVIs | Healthy | 36 | 14 | 50 | 72% | | |
| | Infected | 15 | 32 | 47 | 68.1% | | |
| | Sum | 51 | 46 | 97 | | 70.1% | 0.401 |
| | PA | 70.6% | 69.6% | | | | |
| VIs + MFs | Healthy | 38 | 12 | 50 | 76% | | |
| | Infected | 13 | 34 | 47 | 72.3% | | |
| | Sum | 51 | 46 | 97 | | 74.2% | 0.484 |
| | PA | 74.5% | 73.9% | | | | |
| PIVIs + MFs | Healthy | 42 | 9 | 51 | 82.4% | | |
| | Infected | 9 | 37 | 46 | 80.4% | | |
| | Sum | 51 | 46 | 97 | | 81.4% | 0.628 |
| | PA | 82.4% | 80.4% | | | | |

The above results indicate that the proposed RF model using phenological information-based vegetation indices combined with meteorological features can be used for wheat stripe rust prediction.

In addition, the occurrence of wheat stripe rust in the study area was predicted using the optimal prediction model, i.e., the RF prediction model using PIVIs + MFs (Figure 9). The results showed that the northern border had a heavier incidence than the southern border of the study site. It has been shown that the incidence of stripe rust is generally higher in the late developmental stage than in the early stage [77], and our phenological

prediction results showed that the northern developmental stage occurred earlier than the southern developmental stage (Figure 5). In addition, the occurrence of stripe rust is relatively sporadic in most regions. In western Guanzhong, China, stripe rust is generally at the sporadic stage in mid-April, which can lead to an epidemic of the disease and cause extensive damage if prevention is not timely [70]. The predicted results are consistent with the epidemiological pattern of wheat stripe rust and our field survey. These results demonstrate the feasibility of our proposed stripe rust prediction method, which can provide timely and accurate guidance recommendations to plant protection departments.

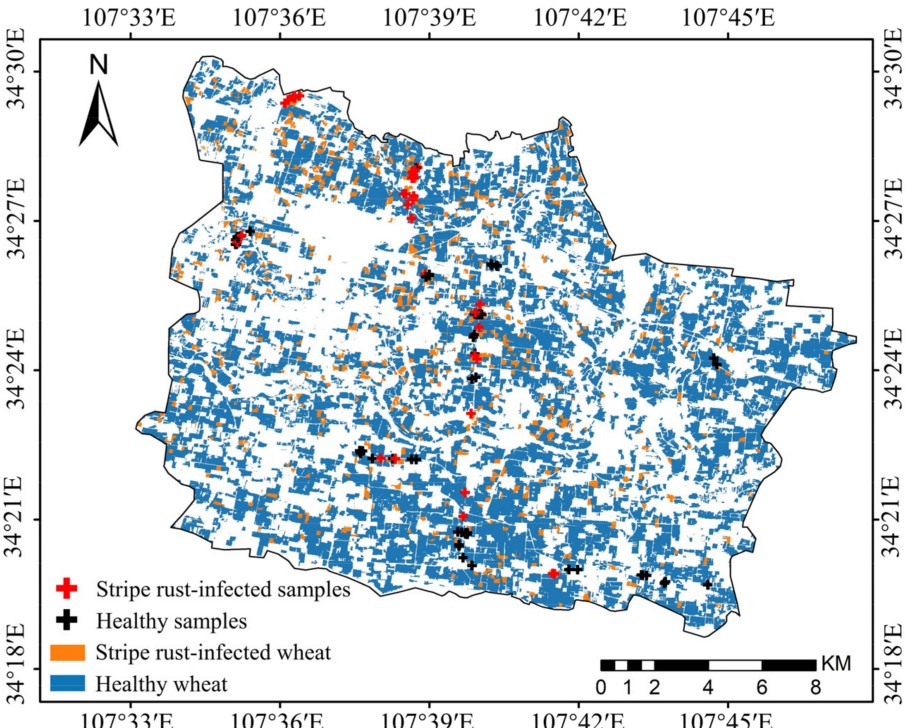

**Figure 9.** Prediction and mapping of wheat stripe rust in Qishan County based on the random forest (RF) model using phenological information-based vegetation indices combined with meteorological features (PIVIs + MFs). Orange represents stripe-rust-infested wheat, blue represents healthy wheat, red cross represents stripe-rust-infected samples and black cross represents healthy samples.

### 3.4. Future Work

Although this study demonstrated the feasibility of predicting wheat stripe rust using phenological information-based vegetation indices combined with meteorological features and satisfactory results were obtained in predicting wheat stripe rust, there are still some shortcomings that need to be improved in future studies. First, although predicting disease distribution at the jointing stage can effectively prevent the spread of the disease, stripe rust occurs throughout multiple phenological stages of wheat, and the disease severity increases gradually under suitable conditions [9,78]. Time-series prediction of disease severity can provide stage guidance for accurate disease prevention and control [15]. Subsequent studies should consider collecting data at different phenological stages on a large scale to validate the potential of the phenological information-based vegetation indices for time-series prediction of wheat stripe rust severity. Second, the growth and habitat conditions sensitive to stripe rust differ at different phenological stages [2,3,12]. The next step will be to analyze the factors sensitive to stripe rust at different periods and to construct a wheat stripe rust time-series prediction model using vegetation indices combined with meteorological features. Third, in addition to the wheat growth and habitat conditions that were considered in our methods, numerous other factors influence the occurrence of wheat stripe rust, and the factors cannot be ignored under certain circumstances. For

example, fertilizer, soil types, humus, mineral composition, other pests and diseases are also important, especially nitrogen fertilization, which has been proven to be influential in wheat stripe rust occurrence [79]. In future studies, we will design a field survey under different stresses to analyze the effects of these factors on stripe rust and collect this information to enhance our model. Fourth, although meteorological interpolation data were validated using the leave-one-out method and acceptable results were obtained, the number of weather stations was small. In order to obtain more accurate meteorological data, we will collect more intensive weather stations (for example, the national automatic stations with a denser distribution of stations than national benchmark weather stations) to verify and improve the prediction model. Finally, the dataset size of wheat stripe rust in this study was small due to the influence of weather and manpower. In order to extend our method to a larger area, we will collect wheat stripe rust data from large areas and collect these biotic and abiotic factors to develop a method for time-series prediction of stripe rust under different stress conditions.

## 4. Conclusions

In this study, a prediction model for wheat stripe rust using remote sensing and meteorological data was developed by combining vegetation indices with meteorological features. The study extracted six phenological information-based vegetation indices (PSRI, NDVI, DSWI, TVI, REDSI and NDVIre) and four meteorological features (RHU_M12, PRE_M01, SSD_M01 and PRE_B07) as input features and constructed a wheat stripe rust prediction model based on RF, SVM and logistic methods. The results showed that the accuracy of the prediction model using PIVIs was better than that using VIs, with an accuracy improvement of 7–16%. In addition, combined with the meteorological characteristics closely related to stripe rust, the accuracy of the prediction model was improved by 9–12%. The RF prediction models outperformed the SVM and logistic prediction models. Among them, the RF prediction model using PIVIs + MFs was the best among all models, with an overall accuracy and kappa coefficient of 88.7% and 0.772, respectively. These results demonstrate that a combination of phenological information-based vegetation indices and meteorological features can be used to predict wheat stripe rust. Our method improves the prediction accuracy of wheat stripe rust compared with that of traditional prediction methods. The proposed method is a feasible solution for predicting wheat stripe rust and provides early warning information to farmers and plant protection departments. In the future, we will explore the time-series prediction method for wheat stripe rust on a large scale to improve the stability and frequency of the prediction model.

**Author Contributions:** Conceptualization, C.R. and Y.D.; formal analysis, Y.D., W.H. and H.M.; funding acquisition, Y.D. and W.H.; investigation, L.H., A.G. and R.S.; methodology, C.R. and W.H.; writing—original draft, C.R.; writing—review and editing, C.R., Y.D., W.H. and H.Y. All authors have read and agreed to the published version of the manuscript.

**Funding:** This work was supported by National Key R&D Program of China (2021YFE0194800), Beijing Nova Program of Science and Technology (Z191100001119089), Alliance of International Science Organizations (Grant No. ANSO-CR-KP-2021-06), Program of Bureau of International Cooperation, Chinese Academy of Sciences (183611KYSB20200080), SINO-EU, Dragon 5 proposal (ID 57457).

**Data Availability Statement:** Data sharing is not applicable to this article.

**Conflicts of Interest:** The authors declare no conflict of interest.

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
