# Peer review of "Integrating Remote Sensing and Meteorological Data to Predict Wheat Stripe Rust"

_remotesensing, doi:10.3390/rs14051221_

Round 1
Reviewer 1 Report
In the manuscript, the authors presented the results of applying machine learning methods to assess disease risk in wheat. Conceptually, the work represents an improvement on existing approaches by using higher resolution remote sensing data (Sentinel 2 instead of Landsat 8). As the authors write, the key difference in improving recognition quality was the use of phenological stages as a decoding feature. There are several global questions about the essence of the work performed. 1. Lines 130-137 - how many points were used to interpolate meteorological data? what method was used to interpolate? what is the accuracy of interpolation on an independent dataset? This is critical information, since, in fact, the meteorological factor is the only factor the authors consider. The reviewer has doubts about the authors' ability to achieve a correct 10 meter raster resolution, since you need at least 1 meteorological station in each field for a more or less equivalent order of accuracy. The authors did not present the results of interpolation in the form of plots-drawings, which also raises questions. 2. The occurrence of rust on wheat is primarily influenced by improper application of nitrogen fertilizer. The authors do not consider this factor either as nitrogen content in the soil or as a map of fields, which also raises questions when evaluating the results obtained. 3. Soil types, humus, and mineral composition are not taken into account at all when assessing wheat disease risk, just like geomorphological conditions and moisture potential of the fields. 4. Phenological phases of plant growth primarily reflect the achievement of a certain state of vegetation growth, including biomass recruitment. The highest biomass of plants reaches already after passing the peak of greenery, which means that the authors incorrectly defined phenological dates. What in this case the authors determined with regard to the vegetation indices is a big question, to which I did not find an answer in the manuscript. In view of all of the above, I cannot recommend the manuscript for publication.Author Response
Dear Reviewer,
We really appreciate your suggestions and comments. We agree with these suggestions and have significantly revised the manuscript accordingly.
We have addressed a point-by-point response to the questions and comments. Please see the content of the attachment “Response to Reviewer”.

Reviewer 2 Report
I appreciate this paper. It is well written, concise, scientifically sound, with great potential to support decision-making processes. The Introduction gives a broad overview of the problem tackled and achievements to date. The Materials and Methods section describes both the data used and the methodology used comprehensively. The results are significant and a discussion with the published articles shows their importance and validity. The only small suggestion concerns lines 264-271 which should be moved to the second part as they concern methodology.
Author Response
Dear Reviewer,
We really appreciate your suggestions and comments. We agree with these suggestions and have significantly revised the manuscript accordingly.
We have addressed a point-by-point response to the questions and comments. Please see the content of the attachment “Response to Reviewer”.

Reviewer 3 Report
I have read with interest the manuscript by Ruan et al. (2022). The analysis is comprehensive, the material is presented concisely and the analysis seems to be technically correct. The application is meaningful and is supported by data collected in the field. The results are certainly of interest. I have some minor comment that may improve the quality of the presentation.
- The choice of the classification algorithm (support vector machine) seems a bit outdated. Simpler and more accurate algorithms (e.g. random forests) are used more frequently with success at least in the last five years.
- It is not clear how the results are transferable to other areas given the local character of the study, and how the proposed approach could be implemented in the national scale, given the need for in situ data.
- Although 10-fold cross-validation is used to assess the predictive performance of the models, the robustness of the results seems a bit questionable given the size of the dataset (please correct me , but the dataset seems to include 97 points to be predicted).
- Furthermore, while the dataset seems to be small, the number of predictor variables seems to be high (please correct me here, but it seems that there are 17 predictor variables in Figure 6). In such cases, I am not sure whether support vector machines work well, unlike e.g. random forests that are suitable for high dimensional problems.
- Finally, I understand the difficulty in collecting in situ data, but problems related to the size of the dataset should be further discussed.
Author Response

(The authors gave the same response as above.)

Round 2
Reviewer 1 Report
The authors slightly adapted the text of the manuscript according to the recommendations, but did not incorporate the explanations given in the response to the reviewer. I suggest that the authors make changes to the manuscript, including in the sections with description with obtaining meteorological factors, vegetation curve analysis, etc.
Author Response
Dear Reviewer,
We really appreciate your suggestions and comments. We agree with these suggestions and have significantly revised the manuscript accordingly.
Our point-by point response and/ or changes to the reviewers’ suggestions/ comments are listed as follows. Please see the attachment “Response to Reviewer”.
